# Immune Environment and Immunotherapy in Endometrial Carcinoma and Cervical Tumors

**DOI:** 10.3390/cancers15072042

**Published:** 2023-03-29

**Authors:** Alexandra Lainé, Andrea M. Gonzalez-Lopez, Uzma Hasan, Ryotaro Ohkuma, Isabelle Ray-Coquard

**Affiliations:** 1Centre Léon Bérard (CLB), 69373 Lyon CEDEX 08, France; 2University Hospital Foundation Alcorcon, 28922 Alcorcon, Madrid, Spain; 3CIRI, Team Enveloped Viruses, Vectors and ImmunotheRapy INSERM U1111/UCBL 1, Centre National de la Recherche Scientifique (CNRS), UMR5308, ENS de Lyon, Université Lyon, 69364 Lyon, France; 4The Lyon Immunotherapy for Cancer Laboratory (LICL), Centre de Recherche en Cancérologie de Lyon (CRCL)/UMR Inserm 1052/CNRS 5286, CLB, 69373 Lyon CEDEX 08, France; 5Team CISTAR, CRCL, INSERM-1052/CNRS-5286, CLB, Lyon, 69373 CEDEX 08, France; 6Division of Medical Oncology, Department of Medicine, School of Medicine, Showa University, Tokyo 142-8555, Japan; 7Centre Léon Bérard, University Claude Bernard Lyon I, 69373 Lyon CEDEX 08, France

**Keywords:** cervical cancers, endometrial cancers, gynecological cancers, immunotherapy, tumor microenvironment

## Abstract

**Simple Summary:**

Endometrial (EC) and cervical cancers (CC) do not respond well to chemotherapy and alternative therapeutic options are lacking. Recently, EC and CC patients’ outcome were improved thanks to immunotherapy-based treatments leading to their approval in the management of these diseases. However, not all patients respond to such treatments, especially in relapsed settings. In the present review, we focused on the immunological features of both EC and CC and presented the upcoming clinical trials targeting the immune system alone or in combination that will allow to improve patient’s care. We also raise some questions that need to be solved in this field and presented the expected progresses.

**Abstract:**

Endometrial cancer (EC) is the seventh most common tumor in women, and prognosis of recurrent and metastatic disease is poor. Cervical cancer (CC) represents the fifth most common gynecological cancer. While ECs are more common in developed countries, the incidence of CC has decreased due to the recent implementation of large screening and vaccination programs. Until very recently, patients with advanced or unresectable EC or CC had very limited treatment options and were receiving in first line setting platinum/taxane-based chemotherapy (CT). Significant progress in the treatment of gynecological cancers has occurred in the last few years, with the use of innovative targeted therapies and immunotherapy. However, targeting the immune system in patients with gynecological tumors remains challenging and is not always successful. In ovarian cancer, several immunotherapy treatment regimens have been investigated (as monotherapy and combination therapy in first and subsequent lines of treatment) and showed poor responses. Therefore, we specifically focused our review on EC and CC for their specific immune-related features and therapeutic results demonstrated with immunotherapy. We report recent and current immunotherapy-based clinical trials and provide a review of emerging data that are likely to impact immunotherapy development based on increased biomarkers’ identification to monitor response and overcome resistance.

## 1. Introduction

Gynecological cancers include ovarian, cervical, endometrial, vaginal, and vulvar cancers that all present specific clinical and biological features. Up to 1,398,601 women were diagnosed with gynecological cancers worldwide in 2020 and over 671,875 related deaths were reported [1] (Table 1). The present review will focus on uterine-associated tumors, i.e., endometrial cancers (EC) (upper part, uterine corpus) and cervical cancers (CC) (lower part, uterine cervix), which showed less pronounced responses to chemotherapy (CT) than ovarian cancers, underlining the need for further therapies including immunotherapy-based treatment, in patients with EC and CC. The use of immunotherapy and in particular immune checkpoint inhibitors (ICIs) drastically changed the management and outcomes of patients, with the ability to boost endogenous immunity against unique tumor antigens, resulting in long-lasting responses in some selected patients. In patients with ovarian cancer, only a limited benefit was achieved, as compared with other tumors [2], and is partly explained because of high tumor heterogeneity and predominant immunosuppressive tumor microenvironment (TME). Several trials are currently investigating the use of ICIs alone or in combination, with high expectations in innovative therapies.

However, results with immunotherapy in patients with EC and CC are still disappointing, especially in the second line setting. Therefore, we reviewed recent achievements and suggested implementations in line with the currently expected progress.

### 1.1. Endometrial Cancers

EC ranks seventh among the most common diagnosed gynecological cancer worldwide and is the fourth most common female cancer in Europe with an incidence of 16.4–20.2/100,000 [1]. Up to 80% of ECs are diagnosed owing to associated postmenopausal bleeding and are mainly locally advanced/confined to the uterus. The mortality rate remains low at 2.7–3.7/100,000 [1,8]. The 5-year overall survival (OS) in patients with stage I EC is over 95% and remains over 50% in EC with invaded lymph nodes but reduced to only 20–25% in patients with advanced/recurrent/metastatic disease [5] (Table 1). Furthermore, EC patients at high and at high–intermediate risk are more prone to relapse with 16–35% relapsing patients compared with patients at intermediate and at low risk relapsing in 9% of the cases [9] (Table 2 and Table 3). Several risk factors have been associated with an increased likelihood of developing EC and a poorer OS including obesity, type 2 diabetes, prior breast or ovarian cancer, and advanced age (Figure 1) [10,11,12,13,14].

In the first-line setting, treatment regimen is based on hysterectomy, with or without adjuvant radiotherapy (RT), or taxane-platinum CT, or hormonotherapy. In advanced or recurrent EC not eligible for tumor resection, no established standard of care in the second-line setting currently exists. CT in this patient population has limited success, and its use particularly faces constraints related to fertility preservation.

The clinical trials Keynote-158 and -146, and more specifically the phase III Keynote-775 trials, prompted the Food and Drug Administration (FDA), in 2019, to approve the combination of the anti-PD-1 pembrolizumab (Keytruda^®^) with the tyrosine kinase inhibitor lenvatinib (Lenvima) in patients with progressive EC after prior systemic therapy and patients not eligible for RT or curative surgery not identified with tumors with microsatellite instability hypermutated (MSI-H) or with deficient mismatch repair (dMMR) (Figure 1) [15,16,17,18]. The European Medicines Agency (EMA) approved, in 2021, the combined therapy in patients with progressive EC during or following CT and patients not eligible for curative surgery or RT. The GARNET trial investigating dostarlimab (Jemperli) in patients not responding to CT led the FDA and the EMA to approve dostarlimab as a monotherapy in 2021 in patients with MSI-H/dMMR tumors (Figure 1) [19,20,21], and further progress in the management of patients with EC is expected with innovative treatments. However, the prognosis of these patients with advanced/metastatic, progressive, or relapsing EC after at least one prior systemic treatment is poor. Hence, the development of new innovative treatments and better identification of the patients who are more likely to respond to dedicated treatments should be encouraged.
cancers-15-02042-t003_Table 3Table 3FIGO staging for endometrial cancers (Koskas et al., 2021 [22]). FIGO, International Federation of Gynecology and Obstetrics grading; M, metastasis; N, lymph nodes; T, tumor; x, any number.FIGO StageTNM CategoryDescriptionIT1N0M0Tumor confined to the uterus, including endocervical glandular involvementIAT1aN0M0Tumor confined to the endometrium or invades <50% of the myometrial wallIBT1bN0M0Invasion ≥ 50% of the myometriumIIT2N0M0Tumor invades cervical stroma, but does not extend beyond the uterusIIIT3N0-1M0Local and/or regional spread of the tumorIIIAT3aN0M0Tumor involves serosa or adnexa (direct extension or metastasis)IIIBT3bN0M0Tumor involves vagina or parametrial (direct extension or metastasis)IIICT1-3N1M0Tumor with pelvic and/or para-aortic lymph node metastasisIIIC1T1-3N1M0Tumor with positive pelvic nodesIIIC2T1-3N1M0Tumor with positive para-aortic nodes with or without positive pelvic lymph nodesIV
Tumor invades bladder and/or bowel mucosa and/or involves distant organsIVAT4NxM0Tumor invades bladder mucosa and/or bowel mucosaIVBTxNxM1Distant metastasis, including intra-abdominal metastasis and/or inguinal nodes


### 1.2. Cervical Cancers

Cervical cancer (CC) is the fifth most frequent and a fatal cancer in women and ranks the ninth most common cancer diagnosed worldwide [1] (Table 1). In Europe, CC is the fourth most common female cancers with an incidence rate of 7.0–14.5/100,000 cases, and the mortality rate is low at 2.0–6.1/100,000 cases [1]. However, the incidence rate greatly varies according to socioeconomic status and a higher incidence (18.8% vs. 11.3%) and mortality (12.4% vs. 5.2%) is reported in low/medium economically developed countries [1,23]. Currently, thanks to advances in the understanding of the aetiopathogenesis of CC and its natural history, these tumors are considered to be a sexually transmitted disease, associated with chronic infection by various genotypes of the human papillomavirus (HPV) (Figure 2). Prior infection with HPV has been reported as a major risk factor for CC development, with 96% of the CC identified in HPV-positive patients [24,25]. More than 150 HPV genotypes have been identified, with type 16 and 18 HPV classified as the highest risk. The implementation of preventive HPV vaccination programs and large screening strategies has contributed to reduce CC incidence and mortality in developed countries [23,25]. Other risk factors for CC development include sexually transmittable co-infections, such as Human Immunodeficiency Virus (HIV), *Chlamydia trachomatis,* and genital Herpes Simplex Virus (HSV), but also tobacco, early age of sexual debut, multiple sexual partners, a high number of childbirth, long-term use of oral contraceptives, and age (Figure 2) [1,26,27,28]. Interestingly, seropositivity to HSV type 2, but not type 1, was shown to be associated with CC in a cohort of 8814 women [29]. The prevalence of HPV infection is relatively high; however, infection is generally fought off by the immune system. Infected individuals rarely develop chronic infections. The carcinogenic mechanism is mainly related to an increased expression of the E6 and E7 viral genes. The related proteins, E6 and E7, inactivate p53 and the retinoblastoma (Rb) gene, respectively, which are central in cell cycle regulation [30]. Their inactivation leads to genomic instability which translates into an appearance of dysplasia with a peak of incidence maximum at 10 years after viral exposition, and neoplastic progression may occur 15 to 20 years later. Cell-mediated immunity via antigen-presenting cells (APCs) is responsible for the activation of the immune response against infections and contribute to the clearance or persistence of the infectious process [31]. However, several ways of evading the immune system exist such as the increased PD-L1 expression, and the downregulation of the major histocompatibility complex (MHC) on the surface of APCs induced by the interaction with the viral protein E5 (Figure 2) [32]. Therefore, this interaction between the virus and the immune system supports the rationale for immunotherapy and therapeutic vaccines in CC.

Prognostic factors for CC include the stage of the tumor, lymph node status, and histology [33] and disparities are found according to race/ethnicity [34]. Cervical adenocarcinoma and adenosquamous carcinoma, accounting for 20% and 4% of all CCs, respectively, are generally associated with poorer prognosis in patients with locally advanced stages (defined according to the International Federation of Gynecology and Obstetrics grading (FIGO) II/III (Table 4)) when compared with the major representative class of squamous cell carcinoma, which accounts for 70% of CCs. Importantly, in patients having received concurrent chemoradiotherapy (CRT), the worst survival rate is reported in patients with cervical adenocarcinoma compared with adenosquamous patients (HR 1.14, 95%CI 1.03–1.27). Considering lymph node involvement, the 5-year survival rate in stages IB–IIA CC with and without lymph node involvement are, respectively, 51–78% and 88–95% (Table 1) [4,35]. More advanced stages showed more reduced survival rates.

The treatment of CC patients varies according to the extent of the tumor. In early stages, conization or hysterectomy are recommended [36]. Adjuvant RT or CRT is considered in the presence of combined risk factors after primary radical surgery when lymph nodes are invaded and positive surgical margins are detected. In locally advanced tumors, the combination of CRT with cisplatin or carboplatin remains the standard of care in patients with a poorer health status [37]. Interestingly, in advanced stages, the phase III trial GOG-204 did not show differences in patients treated with cisplatin combined with paclitaxel, topotecan, gemcitabine, or vinorelbine [38]. Considering the high overall response rate (ORR) (29.1%), median progression-free survival (PFS) (5.8 months), median OS (12.8 months), and a favorable toxicity profile, the combination of cisplatin with paclitaxel is the preferred option in the first-line setting in patients with advanced/metastatic CC not eligible for surgery or RT. The addition of bevacizumab to standard CT is recommended for patients with a good performance status and recurrent, persistent, or metastatic CC [36,39,40,41]. Nevertheless, their outcomes remain poor with a 5-year recurrence rate between 28 to 73.6% for stages IIB-IVB, and treatment options are very limited [42]. Moreover, several side effects are reported with CT, especially in elderly patients, and no alternative therapy after two lines of CT is available.

Some recent advances in immunotherapy, resulting from the trials Keynote-826 and EMPOWER-Cervical 1/GOG-3016/ENGOT-cx9, led to the approval of two new drugs: pembrolizumab and the anti-PD-1 cemipilimab (Libtayo) (Figure 2) [43,44]. The FDA approved pembrolizumab in the first-line alone or in association with CT, with or without bevacizumab, in PD-L1 positive CC with persistent, recurrent, or first-line metastatic disease and who had never received CT [45]. Both the FDA and EMA approved cemipilimab in patients with persistent, recurrent, or metastatic CC regardless of their PD-L1 status, and in case of disease progression during or after platinum-based CT [46]. Other trials are underway to further investigate new therapeutic strategies based on the recently demonstrated sensitivity of CC to immunotherapy and to identify biomarkers predictive for a treatment response.

## 2. The Immune Microenvironment of Endometrial Cancers and Immunotherapy-Based Clinical Trials

The TME includes immune cells, stromal cells, blood cells, and extracellular matrix, playing an active role in tumor development [47]. TME composition differs between tumor types. Innate and adaptive mechanisms are continuously involved in eliminating cancer cells. Therapeutic agents, ICIs or other molecules, can induce and potentiate immunosurveillance through increased tumor antigen (Ag) presentation and T-cell activity, or target the TME through the inhibition of immune mechanisms that promote tumor growth. In several cancer types, immunotherapy has improved patients’ survival [48,49,50,51]. With the exception of platinum-sensitivity, gynecological cancers do not substantially benefit from usual CT regimens. Despite the emergence of new therapies, both incidence and mortality are increasing in patients with EC. As recent advances in the management of EC disease arose from immunotherapy, new therapeutic strategies targeting the immune system need to be identified to improve patients’ survival and better adapt treatment according to tumor phenotypes. To this end, better identification of biomarkers to monitor treatment response is required. We provide an overview of the immune microenvironment in EC and present the main trials investigating immunotherapy and their respective objectives.

### 2.1. Immune Environment of Endometrial Cancers

Endometrial tissue contains numerous leukocytes varying in number and phenotype throughout the menstrual cycle [52]. Leukocytes are more abundant before menstruation, probably in relation to the immune protection required during endometrial disruption. Therefore, tumor immune response may be specifically enhanced in EC cells. A high expression of immune checkpoints (ICP) and their ligands have been reported in ECs, but expression levels vary according to studies and patient cohorts, tumor grade, histology, or mutation status. EC can be classified into four distinct molecular subtypes linked to clinicopathological features [5]: (i) pathogenic polymerase ε (POLE) variants (POLEmut) [53], (ii) dMMR or MSI-H tumors, (iii) p53-mutant (p53-mut), and (iv) no specific molecular profile (NSMP) (Table 2). About 30% of ECs are MSI-H [54]. Therefore, EC represents the most frequent cancer with this dMMR status.

The POLEmut and the MSI-H subtypes show a high mutational load and are thus associated with a higher immunogenic profile, with more tumor-specific neo-Ag resulting in CD4^pos^ and CD8^pos^ tumor infiltration. Several studies have showed that these subtypes are characterized by an increase of PD-1^pos^, PD-L1^pos^ and CD8^pos^ T cells, activated CD8^pos^ T cells (GranzymeB^pos^), and overexpress genes involved in their cytotoxic functions (T-bet, Eomes, interferon γ (IFN-γ), perforin) and exhaustion (LAG-3, Tim-3, TIGIT) [55,56,57,58]. Regardless of EC molecular subtype, the number of intratumoral CD8^pos^ T cells correlates with a better OS and CD8^pos^/FOXP3^pos^ ratio with disease-free survival [59,60]. Interestingly, Tumeh et al., suggested that tumor regression following anti-PD1 therapy requires pre-existing CD8^pos^ T cells [61]. However, the presence of FOXP3^pos^ regulatory T cells (Tregs) impacts the prognosis of patients but not OS [62]. In line with the approval of pembrolizumab for all EC and dostarlimab for MSI-H EC, the MSI-H status may be used as a biomarker of response to ICIs [63]. A recent meta-analysis showed that PD-L1 expression is significantly associated with advanced tumor stages and that PD-L1 overexpression in immune cells, but not tumor cells, correlates with poorer survival, suggesting its use as a biomarker to stratify patients eligible for anti-PD1/L1 therapy [64]. Therefore, other therapeutic options and biomarkers need to be provided in patients with tumors not sensitive to anti-PD-1 treatment or MSS.

The immune modulatory molecule indoleamine 2,3-dioxygenase 1 (IDO1) involved in EC, converts L-tryptophan into the immunosuppressive metabolite L-kynurenine, that can promote Tregs cell differentiation, induce tolerogenic dendritic cells (DC), and myeloid-derived suppressor cells (MDSC) (Figure 2) [65]. High levels of IDO1 in EC are associated with disease progression and impaired clinical outcome [66]. IDO (1 and/or 2) expression in EC correlates with the number of nodal metastases, low number of CD8^pos^ T cells and natural killer (NK) cells within the tumor, and decreased PFS [66,67,68]. In 2018, it was shown that the majority of EC cells that express IDO also express PD-L1 [69]. IDO is mostly expressed in MSI-H EC but also in a subset of proficient MMR (pMMR) patients, highlighting that anti-IDO therapy may be an option for non MSI-H EC patients.

Lymphocyte activation gene-3 (LAG-3), also known as CD223, is an ICP participating in the immune escape of tumors. LAG-3 is expressed by activated T cells, NK cells, B cells, and DC and binds to the MHC class II (MHC-II) or to the galectin-3 (Gal-3) expressed by tumor cells. The binding of LAG-3 and its ligands increase intratumoral T cells, the inhibition of CD4^pos^ T cell activation and CD8^pos^ T cell cytotoxic functions, and tumor cell evasion of apoptosis [70,71,72]. In EC, around 30% of the tumor cells and 25% of the immune cells express this ICP with a prevalent expression in POLEmut and dMMR tumors [71,72]. Interestingly, LAG-3 expression is correlated with Gal-3 in EC, predominantly in dMMR EC tumors [71]. The blockade of LAG-3, therefore, represents an interesting strategy in EC, but a better characterization of the population more likely to benefit from this therapeutic approach needs to be clearly defined. In addition, the inhibition of the CD8^pos^ T cells’ antitumor response differs from LAG-3 and PD-1 related responses, suggesting that a combined inhibitory strategy could enhance antitumor immune responses in EC. In line with this hypothesis, the RELATIVITY-047 phase II/III clinical trial demonstrated a benefit of anti-LAG-3 combined with anti-PD-1 in melanoma patients, allowing the FDA’s approval of the combined treatment with relatlimab and nivolumab [73].

Altogether, these findings laid the groundwork to further investigate immunotherapy in EC. Therefore, several trials have been carried out to evaluate the efficacy of ICIs, alone or combined to other targeted therapeutics, but detailed mechanisms of action still need to be precised and a better identification of eligible patients needs to be performed.

### 2.2. Immunotherapy in Endometrial Cancers: Recent and Ongoing Trials (Table 5)

Recent investigations with immunotherapy, especially anti-PD1/L1, have been conducted in patients with EC, and response is associated with higher rates of tumor-infiltrating lymphocytes (TILs) in patients with MSI-H EC. Herein, we will present recently completed and ongoing trials investigating immunotherapy alone or in combination in EC. Other studies investigating targeted therapies such as the cyclin-dependent PI3K, WEE1, and HER2 inhibitors have been performed; however, these approaches are not presented here.

**Table 5 cancers-15-02042-t005:** Ongoing clinical trials testing immunotherapies alone or in combination in endometrial cancer. DC, dendritic cells; EC, endometrial cancer; FGFR, fibroblast growth factor receptor; FRα, folate receptor α; HER2, Human Epidermal Receptor-2; IDO1, Indoleamine 2,3-dioxygenase; MDSC, myeloid-derived suppressor cells; ORR, overall response rate; OS, overall survival; PARPi, Poly(ADP-ribose) polymerase inhibitor; PFS, progression-free survival; TAM, tumor-associated macrophages.

Clinical Trial	Phase	Condition or Disease	Number of Patients	Drugs Combination	Mechanism of Action	Primary Endpoint (Time Frame)	Status
**First-line treatment**							
DOMENICA (NCT05201547) [74]	III	dMMR (IIIC2/IV or first recurrent EC without curative treatment by RT, CT or surgery)	142	Dostarlimab vs. Chemotherapy	Anti-PD-1 (dostarlimab)	PFS (5 years)	Recruiting
AtTEND (NCT03603184) [75]	III	EC with residual disease after surgery or inoperable FIGO III–IV	550	Atezolizumab or placebo + taxane platinum-based CT	Anti-PD-L1 (atezolizumab)	OS and PFS (2 years)	Active, not recruiting
RUBY (NCT03981796) [76]	III	FIGO III-IV or first recurrent EC	785	Dostarlimab (or placebo) + taxane platinum-based CT followed by dostarlimab or niraparib (or placebo)	Anti-PD-1 (dostarlimab) and PARPi (niraparib)	Investigator assessed PFS and OS (6 years)	Active, not recruiting
EnGOT-en9 (NCT03884101) [77]	III	First-line treatment of FIGO stage III, IVA, IVB or recurrent EC	875	Pembrolizumab+ lenvatinib vs. taxane platinum-based CT	Anti-PD-1 (pembrolizumab) and protein kinase inhibitor (lenvatinib)	PFS (31 months) and OS (45 months)	Active, not recruiting
**Advanced and recurrent EC**							
NRG-GY018 (NCT03914612) [78]	III	FIGO stage III, IVA, IVB or recurrent EC	810	Pembrolizumab or placebo + taxane platinum-based CT	Anti-PD-1 (pembrolizumab)	PFS (5 years)	Active, not recruiting
GYNET (NCT04652076) [79]	I/II	Pretreated advanced EC	240	Netrin-1 mAbs (NP137) + Carboplatin Plus Paclitaxel and/or Pembrolizumab	Anti-PD-1 (pembrolizumab) and anti-netrin-1 (NP137)	Dose limiting toxicity occurrence, ORR (12 weeks up to 2 years)	Recruiting
NCT04885413 [80]	II	Recurrent/advanced EC	37	Niraparib + sintilimab	PARPi (niraparib) and anti-PD-1 (sintilimab)	ORR (6 months)	Recruiting
NCT02912572 [81]	II	Recurrent or persistent EC MSI-H and/or POLEmut, MSS	105	Avelumab or Avelumab and Talazoparib or Avelumab and Axitinib	Anti-PD-L1 (avelumab) and anti-HER2 (talazoparib) and tyrosine kinase inhibitor (axitinib)	Activity of avelumab + talazoparib in EC, activity of avelumab + axitinib in MSS patients assessed by frequency of patients with PFS > 6 months or with objective tumor response (2 years)	Recruiting
NCT05036681 [82]	II	Metastatic EC not amenable to surgery or RT MSS	30	Futibatinib and Pembrolizumab	FGFR1-4 (futibatinib) and anti-PD-1 (pembrolizumab)	ORR, safety, and tolerability (1 year)	Recruiting
POD1UM-204 (NCT04463771) [83]	II	Advanced or metastatic EC with CT progression	300	Retifanlimab +/− epacadostat or pemigatinib or anti-LAG3 and anti-TIM3	Anti-PD-1 (Retifanlimab) and anti-IDO1 (epacadostat)	ORR, PFS (2.5 years)	Recruiting
NCT05156268 [84]	II	Persistent/recurrent EC (including CS)	25	Pembrolizumab + olaparib	Anti-PD-1 (pembrolizumab) and PARPi (olaparib)	ORR (24 weeks)	Recruiting
NCT03526432 [85]	II	Pretreated advanced, recurrent, or persistent EC	55	Atezolizumab + bevacizumab	Anti-PD-L1 (atezolizumab) and anti-angiogenic (bevacizumab)	ORR (3 years)	Active, not recruiting
NCT03016338 [86]	II	Pretreated advanced/recurrent EC	51	Dostarlimab + niraparib	Anti-PD-1 (dostarlimab) and PARPi (niraparib)	Clinical benefit rate (16 weeks)	Active, not recruiting
DOMEC (NCT03951415) [87]	II	Recurrent or persistent EC	55	Olaparib + Durvalumab	Anti-PD-L1 (durvalumab) and PARPi (olaparib)	PFS (6 months)	Active, not recruiting
DUO-E (NCT04269200) [88]	III	After first-line treatment of advanced/recurrent EC	699	Taxan platinum-based CT and durvalumab followed by placebo vs. durvalumab vs. durvalumab and olaparib	Anti-PD-L1 (durvalumab) and PARPi (olaparib)	PFS (4 years)	Recruiting
**Recurrent/relapsed/advanced EC dMMR and/or MSI-H only**							
CAN-RESPOND (NCT05550558) [89]	II	Recurrent sporadic dMMR EC	43	Camrelizumab + anlotinib	Anti-PD-1 (camrelizumab) and multitarget tyrosine kinase inhibitor (anlotinib)	ORR (24 months)	Not yet recruiting
NCT05419817 [90]	II	Recurrent EC and other solid tumors dMMR post PD1 exposure	30	Pembrolizumab + Sitravatinib	Anti-PD-1 (pembrolizumab) and receptor tyrosine kinases inhibitor (Sitravatinib)	ORR (12 weeks)	Recruiting
NCT05112601 [91]	II	Recurrent dMMR	12	Ipilimumab + Nivolumab vs. nivolumab	Anti-CTLA-4 (ipilimumab) and anti-PD-1 (nivolumab)	PFS (5 years)	Recruiting
**Recurrent/relapsed/advanced EC MSS only**							
IMGN853 (NCT03835819) [92]	II	Advanced or recurrent serous EC, MSS, FRα^pos^	35	Mirvetuximab Soravtansine (IMGN853) and Pembrolizumab	Anti-FRα (Mirvetuximab soravtansine) and anti-PD-1 (pembrolizumab)	ORR, PFS (6 months)	Recruiting

#### 2.2.1. Anti-PD1/PD-L1

The rationale for the use of antibodies blocking PD-1 or PD-L1 is supported by results from several trials that have tested their efficacy to improve treatment response, or better identify specific drugs likely to benefit dedicated patients, based on efficacy and tolerance. The phase II study JapicCTI-163212 enrolled 64 patients with EC, CC, and soft-tissue sarcoma and tested the clinical efficacy of the anti-PD-1 nivolumab [93]. Patients with EC presented an ORR of 22.7%, a median PFS of 3.4 months, and a 12-month OS of 48.5%. No differences in PD-L1 expression were observed in EC patients. EC patients with the MSI-H phenotype all achieved partial responses. This study, therefore, suggests that MSI-H status may be used as a predictive factor to select patients more likely to respond to nivolumab. Whether the anti-PD-1 nivolumab can be used such as pembrolizumab to treat a patient with EC still needs to be evidenced.

In pretreated patients, the anti-PD-L1 atezolizumab showed an ORR of 13% in a phase I study (NCT01375842) [94]. A phase II study showed that the PD-L1 inhibitor avelumab exhibited promising activity in dMMR EC patients, regardless of PD-L1 level, highlighting that molecular characterization is more useful than PD-L1 expression analysis for patient selection [95]. The phase II trial PHAEDRA (ANZGOG1601) enrolled 71 patients with advanced EC, including 36 with dMMR and 35 with pMMR [96]. In this study, the anti-PD-L1 durvalumab showed promising activity and manageable safety in dMMR patients regardless of prior lines of CT.

The phase III trial RUBY (NCT03981796) recently reported, for the first time, a significant clinical benefit with dostarlimab combined with standard-of-care CT in patients with FIGO III-IV or first recurrent EC patients [76,97]. Moreover, a favorable trend in survival was observed in all patient regardless of molecular subtype (dMMR/MSI-H and pMMR/MSS).

Several studies are currently investigating ICI in monotherapy. First, the phase III trial DOMENICA (NCT05201547) aims to assess PFS with dostarlimab vs. CT in 142 dMMR/MSI-H patients with FIGO stages IIIC2/IV in first-line advanced or first recurrence settings [74]. Secondly, the trial NRG-GY018 (NCT03914612) investigates the efficacy of pembrolizumab alone, in comparison to taxane platinum-based CT in 810 patients who have not previously received first-line CT [78]. In addition, the phase III trial AtTEND (NCT03603184) [75] has planned to enroll, in the first-line setting, 550 EC patients with residual diseases after surgery or inoperable FIGO III-IV stages and to allocate them to receive atezolizumab versus a placebo combined with platinum-based doublet and to assess whether atezolizumab combined with CT is beneficial to patients.

#### 2.2.2. ICI and Anti-Angiogenic/Tyrosine Kinase Inhibitors

Lenvatininb is a kinase inhibitor with anti-angiogenic features already approved in the treatment of EC patients, along with pembrolizumab. Several trials have investigated the combination of these inhibitors with ICI. The phase II trial NCT03367741 showed that the combination of nivolumab and cabozantinib is beneficial to patients with recurrent ECs, treated with 3 or more prior regimens [98]. The inhibitor of multiple tyrosine kinases cabozantinib targeting the Met/Hepatocyte Growth Factor Receptor (HGFR), the AXL receptor tyrosine kinase, and the Vascular Endothelial Growth Factor Receptor 2 (VEGFR-2) showed a median PFS of 5.3 months with the combination therapy vs. 1.9 months with cabozantinib alone. Among patients treated with immunotherapy, non-progressive patients had proportions of activated tissue-resident γδ T cells significantly higher than progressive patients. The combination of these inhibitors with immunotherapy may be beneficial to patients having received prior treatments. A phase II trial (NCT02912572) is currently evaluating the combination of avelumab with the multikinase inhibitor of VEGFR-1, -2, and -3 and the Platelet Derived Growth Factor Receptor (PDGRF) axitinib in EC patients with a recurrent or persistent disease. The phase III trial EnGOT-en9/LEAP-001 (NCT03884101) has planned to enroll 875 patients with FIGO stage III, IVA, IVB, or recurrent EC and will evaluate PFS and OS in first-line treatment with pembrolizumab and lenvatinib vs. taxane platinum-based CT [77]. Interestingly, the phase II trial CAN-RESPOND aims to evaluate the ORR in 43 patients with a relapsing dMMR EC treated with anti-PD-1 camrelizumab combined with the multitarget tyrosine kinase inhibitor anlotinib [89]. In the patients having received prior anti-PD-1 therapy, the phase II trial (NCT05419817) will assess the ORR after the administration of pembrolizumab along with the tyrosine kinase inhibitor sitravatinib [90].

#### 2.2.3. ICI and Polymerase Inhibitors

The outcomes and safety of the combination of avelumab with the polyadenosine diphosphate-ribose polymerase (PARP) inhibitor (PARPi) talazoparib were assessed in a phase II trial that enrolled 35 recurrent pMMR EC patients [81]. Four patients had partial responses and eight were still progression-free at 6 months. While homologous recombination deficiency (HRD) was associated with clinical benefits and better PFS, tumor mutation burden, TILs, and PD-1 status were not identified as predictive factors. Post-hoc analyses showed that patients treated with this combination, who were at a progression-free interval (PFI) at 6 months or longer had clinical benefits. Avelumab with talazoparib can be further considered for recurrent pMMR patients with EC.

The phase II trial (NCT02912572) is currently recruiting patients to evaluate the efficacy of the combinations (i) avelumab and talazoparib or (ii) avelumab with axitinib in recurrent/persistent EC or MSS-recurrent/persistent EC patients, respectively. The trial assessed the rate of patients with a PFS above 6 months or with objective tumor response in a time frame of 2 years. This trial will also evaluate the duration of OS, PFS, and immune-related objective response, giving insights into the rationale of using this combination in MSS patients in need for new treatment options and better identifying and characterizing patients in failure with this regimen.

The phase III trial RUBY (NCT03981796), which has already reported clear benefits for EC treated with dostarlimab and taxane platinum-based CT, will evaluate maintenance with dostarlimab or niraparib [76]. Another ongoing phase III trial is DUO-E (NCT04269200), which will evaluate, on 699 patients with advanced/recurrent EC, the PFS after the combination of durvalumab with olaparib in the first-line regimen [88]. The phase II trial DOMEC (NCT03951415) has already recruited 55 patients with recurrent/persistent EC treated with olaparib and durvalumab [87]. Besides PFS, secondary endpoints include the evaluations of ORR, OS, and the use of tumor biopsies for the quantification of predictive biomarkers including dMMR/POLE status, HRD status, and several immune factors (CD3, CD4, CD8, CD103, PD-1, CD161, LAG3, CTLA-4, NKG2A, FOXP3, NK, and PD-L1) on myeloid and tumor cells, and describe myeloid infiltration (CD68, CD14, CD33, and CD163). Other trials currently evaluating the combination of PARPi with immunotherapy are listed in Table 5.

#### 2.2.4. ICI and Other Targeted Therapies

The innovative phase I/II trial GYNET (NCT04652076) [79] has planned to recruit over 240 patients with CC and EC. The aim is to evaluate the efficacy of restoring apoptosis in tumor cells, using the first-in-class humanized monoclonal antibody that blocks Netrin-1 (NP137) alone or combined with an anti-PD-1 [99]. Netrin-1 is a secreted protein involved in axon guidance, bone remodeling, and metabolic homeostasis. Used as a tumor biomarker, Netrin-1 is highly expressed in cancers, and especially in aggressive cancers. Netrin-1 expression was assumed to favor tumorigenesis, while promoting epithelial to mesenchymal transition (EMT) [100,101].

The phase II trial POD1UM-204 (NCT04463771) is an umbrella study investigating the anti-PD-1 retifanlimab in patients with advanced or metastatic EC who progressed during or after taxane platinum-based CT [83]. Monotherapy will be evaluated in patients with PD-L1^neg^, MSI-H, or dMMR/POLEmut EC and retifanlimab combined with an anti-FGFR2 and will be investigated in patients with mutated ECs or combined with an anti-IDO1 molecule in MSS and PD-L1^pos^ EC patients. The role of IDO-1 in EC was already mentioned. Fibroblast growth factor receptor (FGFR) plays a role in angiogenesis, and FGFR2 mutations found in around 15% of EC leads to FGFR overexpression and favors the proliferation of tumor cells and metastasis [102,103]. Shorter disease-free survival and OS are observed in these patients, especially at an early stage, owing to the lack of treatment options. Therefore, the trial POD1UM-204 will support the management of EC patients through a better identification of EC patients that are potential responders to anti-FGFR2 therapy and will determine if patients with a non MSI-H EC benefit from the inhibition of IDO-1. In MSI-H patients progressive after anti-PD-L1 treatment, the POD1UM-204 trial has proposed an additional treatment with anti-TIM3/LAG-3. OS and PFS will be assessed in all groups of patients. Another phase II (NCT05036681) study will aim at testing the inhibition of FGFR along with anti-PD-1 in metastatic EC patients with an MSS profile who are not candidates for surgery [82].

Other ICI combinations will be evaluated in a subset of EC patients with recurrent dMMR who will receive nivolumab with or without ipilimumab (NCT05112601, phase II trial) [91].

Folate receptors (FR) contribute to cell proliferation via the transportation of folate vitamins across cell membranes. The major subunit FR-α is highly expressed in gynecological cancers. The FR-α expression in EC is significantly higher than in endometrial hyperplasia or normal endometrium [104]. The phase II IMGN853 trial (NCT03835819) will evaluate the ORR and PFS in patients with advanced/recurrent MSS EC and who are FR-α positive, following the administration of the antibody-drug conjugate (ADC) mirvetuximab soravtansine (IMGN853), along with anti-PD-1 [92].

## 3. The Immune Microenvironment of Cervical Cancers and Immunotherapy-Based Clinical Trials

### 3.1. Immune Environment of Cervical Cancers

Almost all the patients with CC are HPV-positive [24]. The most frequent HPV detected in CC samples is HPV16 and HPV18, encoding for E6 and E7 proteins known as tumor drivers by suppressing p53 and Rb genes, respectively, and both participating in immune evasion through an increased PD-L1 expression and induction of an immune tolerant environment [105,106,107]. In addition, inhibition of p53 and Rb preventing normal cell-cycle control and apoptosis leads to the proliferation of HPV-infected damaged cells [30]. HPV infection drives a specific immune response with the proliferation of E6- and E7-specific T cells. However, in CC patients, this immune response is impaired and highlights the need to boost the immune system by ICI combined or not with other targeted therapies.

In 2007, Piersma et al., evaluated the CD8^pos^/FOXP3^pos^ ratio in the CC tissue and blood of patients before a hysterectomy [108]. The authors showed a ratio in patients with localized tumors higher than in metastatic tumors, suggesting its use as a prognostic factor in CC patients. Immunohistochemistry (IHC) analysis in CC tissues—but not in blood samples—from treatment-naïve patients showed that CD8^pos^/FOXP3^pos^ combined with the presence of type 1 macrophages is a significant independent favorable prognostic factor [109,110]. Moreover, a high ratio was associated with more favorable clinical outcomes in patients with CC after neoadjuvant CT [111]. The 5-year survival rate was reported as significantly lower in CC patients presenting a high Treg tumor-infiltration [112].

Another series of 21 patients with CC showed that 11 tumors responding to CT showed an inflammatory status higher than non-responding tumors (N = 10) [113]. Additionally, patients in response presented a higher CD8^pos^ lymphocyte tumor-infiltrate rate, and higher PD-1, -L1, and -L2 levels as well as higher mutation rates in *PIK3CA* and *KDR/VEGFR*. Another study, conducted on 20 patients with CC, reported that tumor samples had decreased CD3^pos^, CD4^pos^, Tregs, and CD8^pos^ T cells during the first week of CRT [114]. After 3 to 5 weeks of treatment, CD4^pos^ and CD8^pos^ T cells presented an activated phenotype. In corresponding blood samples, very few activated T cells were observed.

The presence of the tumor-associated antigens E6 and E7 proteins, the high mutational load, and/or the presence of immune infiltrate, provides new promising approaches to identify predictive favorable factors for a response with immunotherapy. Some studies reported that prior CT may change TME in patients with CC through increased infiltration of several immune-related cells, favoring TME sensitivity to ICI [115]. Targeting the immune system gives rise to new opportunities to improving outcomes in patients with CC.

### 3.2. Immunotherapy in Cervical Cancers: Recent and Ongoing Trials (Table 6)

The standard of care relies on different factors, including tumor extension, and multidisciplinary approaches with RT, CT, and/or surgical resection are usually required. Immunotherapy has drastically changed the treatment of many cancers, and its critical role is also expected for the future management in patients with CC.

**Table 6 cancers-15-02042-t006:** Ongoing clinical trials testing immunotherapies alone or in combination in cervical cancer. CC, cervical cancer; CT, chemotherapy; CRT, chemoradiotherapy; ORR, overall response rate; OS, overall survival; PFS, progression-free survival.

Clinical Trial	Phase	Condition or Disease	Number of Patients	Drugs Combination	Mechanism of Tion	Primary Endpoint (Time Frame)	Status
**First-line treatment**							
NCT05311566 [116]	II	FIGO IB2-IIIB	92	Camrelizumab plus concurrent CRT	Anti-PD-1 (camrelizumab)	OS (3 years)	Recruiting
NCT04974944 [117]	II	FIGO IVB, Recurrent or Persistent	172	Camrelizumab and apatinib vs. paclitaxel and cisplatin and carboplatin and bevacizumab	Anti-PD-1 (camrelizumab) and anti-VEGFR2 (apatinib)	PFS (36 months)	Recruiting
NCT05511623 [118]	II	FIGO IIIC2	112	Tislelizumab and concurrent CRT	Anti-PD-1 (tislelizumab)	PFS and side effects (3 years)	Not yet recruiting
FERMATA (NCT03912415) [119]	III	FIGO IVB	316	BCD-100 + CT +/− bevacizumab	Anti-PD-1 (BCD-100)	OS (3 years)	Recruiting
BEATcc (NCT03556839) [120]	III	FIGO IVB	404	Atezolizumab + CT+ bevacizumab	Anti-PD-L1 (atezolizumab)	PFS and OS (48 months)	Active, not recruiting
NCT04982237 [121]	III	FIGO IVB	440	Cadonilimab + CT +/− bevacizumab	Anti-PD-1/CTLA-4 bispecific antibody (Cadonilimab, AK104)	PFS and OS (2 years)	Recruiting
NCT04973904 [122]	II	Recurrent, Refractory and Metastatic CC	35	Toripalimab + paclitaxel + cisplatin + bevacizumab	Anti-PD-1 (toripalimab)	ORR (1 year)	Not yet recruiting
**Recurrent/persistent/metastatic CC**							
ITTACC (NCT05614453) [123]	II	After Platinum-Based CT	57	Sitravatinib Tislelizumab	Tyrosine kinase inhibitor (sitravatinib) and anti-PD-1 (tislelizumab)	ORR (2 years)	Not yet recruiting
NCT03108495 [124]	II	Recurrent, metastatic, or persistent CC	189	LN-145 and LN-145 + Pembrolizumab	Autologous tumor infiltrating lymphocytes (TIL) infusion (LN-145) followed by IL-2	ORR and AE (60 months)	Recruiting
NCT04646005 [125]	II	Recurrent/Metastatic HPV16 CC	105	Cemiplimab + ISA101b vaccine	Anti-PD-1 (cemipilimab)	ORR (36 months)	Recruiting
NCT05247619 [126]	II	Recurrent, metastatic, or persistent CC	49	Tislelizumab/Bevacizumab/Paclitaxel/Cisplatin/Carboplatin	Anti-PD-1 (tislelizumab)	Median PFS (24 months)	Recruiting
NCT05033132 [127]	II	Advanced CC	177	Balstilimab or balstilimab + Zalifrelimab	Anti-PD-L1 (balstilimab) and anti-CTLA-4 (zalifrelimab)	ORR (36 months)	Recruiting
ALARICE (NCT03826589) [128]	NA	Persistent or recurrent CC after platinum-based CT	23	Avelumab and Axitinib	Anti-PD-L1 (avelumab) and tyrosine kinase inhibitor (axitinib)	ORR (2years)	Recruiting
NCT03808857 [129]	II	Recurrent or metastatic, PD-L1 positive who failed in platinum-based CT	80	GB226	Anti-PD-1 inhibitor (GB226)	ORR (2 years)	Recruiting
STAR (NCT04068753) [130]	II	Recurrent or progressive CC	66	Dostarlimab and niraparib	Anti-PD-1 (dostarlimab) and PARPi (niraparib)	Proportion of patients with response to treatment (1 year)	Recruiting
NCT05446883 [131]	III	First-Line treatment of persistent, recurrent or metastatic CC	498	QL1706 and placebo and paclitaxel and cisplatin/carboplatin with or without bevacizumab	Anti-PD-1 and –CTLA-4 (QL1706)	PFS and OS (2 years)	Recruiting
ATOMICC (NCT03833479) [132]	II	Maintenance therapy for patients with high-risk locally advanced CC	132	TSR-042	Anti-PD-1 (TSR-042)	PFS (30 months)	Recruiting
ATEZOLACC (NCT03612791) [133]	II	Locally advanced CC	189	Atezolizumab and RCT	Anti-PD-L1 (atezolizumab)	PFS (24 months)	Recruiting

#### 3.2.1. PD-1 and PD-L1 Inhibitors

The phase II trial NRG-GY002 (NCT02257528) evaluated the anti-PD-1 nivolumab in 25 patients with persistent or recurrent CC after one prior systemic CT [134]. Although the median OS was 14.5 months, the benefit from nivolumab as a monotherapy was not demonstrated; however, an acceptable safety profile was reported. Among 88% of the patients tested for PD-L1 expression, 77.3% expressed PD-L1 (combined positive score (CPS) cut-off of >1%) but no significant correlation between tumor response and PD-L1 expression was identified. The Checkmate 358 trial enrolled 19 patients with CC, including patients with vaginal/vulvar cancer to receive nivolumab [135]. With a median follow-up of 19.2 months, CC patients reached a median OS of 21.9 (95%CI 15.1-not reached (NR)) months, and the ORR was 26.3% (95%CI 9.1–51.2). The duration of response (DOR) was not reached, and CC patients treated with nivolumab reported a stabilized health status and health-related quality of life. Further trials investigating nivolumab are still required in patients with CC.

Recently, the PD-1 inhibitor balstilimab was investigated in a phase II study in 161 patients with recurrent and/or metastatic CC after one prior line of platinum-based CT [136]. The ORR was 15% (95%CI 10–21.8%) in the overall population, and 20% and 7.9% in PD-L1^pos^ and PD-L1^neg^ patients with CC, respectively. Response to balstilimab was reported as independent of PD-L1 expression and histology, suggesting a potential benefit in a larger population of CC patients, compared with pembrolizumab exclusively recommended in PD-L1^pos^ CC.

The phase III trial CALLA evaluated the combination of CT with durvalumab compared to CT alone in CC patients with FIGO stages IB2 to IVA, naïve of treatment, regardless of lymph node status [137]. Unfortunately, in March 2022, the trial showed no benefit from this combination [138]. However, the GOG-3047 trial will investigate pembrolizumab with CT and RT in a similar cohort of patients [78], and will determine whether CC patients naïve of treatment, may benefit from anti-PD-1 combined with CT. In addition, the phase II trial ATEZOLACC will investigate CRT with or without atezolizumab in patients with locally advanced CC [133]. Finally, the ongoing trial NRG-GY017 is currently evaluating the combination of CT with ICI [139]. With 40 patients enrolled, the study aims to assess the administration of atezolizumab 3 weeks before and/or with CRT in CC patients with positive lymph nodes and FIGO IB2-IVA. The phase II trial ATOMICC will evaluate the role of anti-PD-1 in CC patients with a high-risk type of tumor in maintenance therapy (NCT03833479) [132].

#### 3.2.2. CTLA-4 Inhibitors

The phase II study (NCT01693783) enrolled 42 patients with advanced CC to investigate ipilimumab in the second-line setting [140]. The ORR was 8.3%, and median PFS and median OS were 2.5 and 8.5 months, respectively. The intratumoral expression of CD3, CD4, CD8, FOXP3, IDO, and PD-L1 in pre- and post-treatment samples will be evaluated. More recently, the phase I trial GOG-8829 tested the addition of ipilimumab after standard definitive CRT in 21 CC patients with positive pelvic lymph nodes [141]. The OS and PFS were, respectively, 90% and 81% at one year. In peripheral blood, rates of PD-1^pos^CD4^pos^ and CD8^pos^ T cells after CRT increased, and respective levels were maintained after ipilimumab administration. This study suggests that using immunotherapy after CRT might be a promising and well tolerated approach. Further studies are needed to see if this expression correlates with the tumor and is used as a predictive factor of treatment response.

#### 3.2.3. ICI and Tyrosine Kinase Inhibitors

The multicenter open-label single-arm study CLAP, testing the combination of camrelizumab with the VEGFR-2 tyrosine kinase inhibitor apatinib in patients with advanced CC [142], enrolled 44 patients to receive 200 mg camrelizumab every 2 weeks and 250 mg apatinib once a day and showed an ORR of 55.6% (95%CI 40–70.4%) with 23 partial responses and 2 complete responses. This combination was considered as promising in advanced CC patients, even after prior lines of anti-angiogenic molecules or two or more lines of CT. Larger trials are needed to further investigate these encouraging results as listed in Table 6. In 2023, the tumor tissue collection from the CLAP trial further explored tumor-infiltrating lymphocytes before and after treatment [143]. Multiplex-immunofluorescence showed that CD8^pos^ T cell density in invasive margins in responder patients was higher than in non-responders and was associated with prolonged PFS. In addition, the higher ratio of macrophages (CD68^pos^) to CD8^pos^ T cells observed in non-responder patients was associated with poorer PFS. No significant results were observed with FOXP3^pos^ regulatory T cells. This observation suggests that the localization of tumor-infiltrating CD8^pos^ T cells might be a biomarker of treatment response.

#### 3.2.4. ICI Combination

As reported in patients with other tumors, the combination of two immunotherapy drugs also appears to be a promising strategy in patients with CC. The phase I/II Checkmate-358, testing the combination of nivolumab and ipilimumab, enrolled 19 patients with recurrent or metastatic and virus-associated CC (including 5 patients with vulvar/vaginal cancer) and allocated them by randomization to a 24-month treatment or until progression, or unacceptable toxicity [144]. The primary endpoint, ORR, was higher in the combination arm, especially in patients who had not received any previous treatment. Responses to treatment were independent of PD-L1 status; however, significant toxicity was reported even if manageable. Another study evaluated the combination of balstilimab and the anti-CTLA-4 zalifrelimab in 155 women with recurrent and/or metastatic CC who relapsed after prior platinum-based CT [145]. The primary end-point ORR was 25.6% (95%CI 18.8–33.9) in the overall population and included 10 complete responses. The ORR was 32.8% in patients with PD-L1^pos^ CC and 9.1% in patients with PD-L1^neg^ CC, respectively. This promising combination showed durable clinical activity with a favorable tolerability. The ongoing phase II pilot study COLIBRI (NCT04256213) enrolled 40 patients to assess the CD8^pos^/FOXP3^pos^ ratio in CC patients treated with nivolumab combined with ipilimumab before RCT [146], reporting variations in CD8^pos^/FOXP3^pos^ ratio during treatment and validating its use as a biomarker for treatment response.

The humanized antibody tiragolumab targeting T-cell immunoreceptor with Ig and ITIM domains, TIGIT is an immune checkpoint inhibitor that blocks the binding of TIGIT to its ligand CD155, thereby enhancing T-cell and NK-cell anti-tumor activity. Tiragolumab is currently being investigated in the phase II trial SKYSCRAPER-04 (NCT04300647), comparing atezolizumab with or without tiragolumab in patients with metastatic and/or recurrent PD-L1^pos^ CC [147]. The humanized antibody relatlimab targeting the immune checkpoint lymphocyte-activation gene 3 (LAG-3) is being evaluated in combination with nivolumab in a phase I/II trial in patients with CC among other tumors (NCT01968109) [148].

#### 3.2.5. ICI, Anti-Angiogenics and Other Targets

Besides CT, other combinations involving anti-angiogenic molecules are being evaluated. The addition of bevacizumab to atezolizumab did not improve ORR in patients with advanced CC (NCT02921269) [149]. The study did not meet its primary endpoint, with none of the ten patients enrolled and having received bevacizumab 15 mg/kg and atezolizumab 1200mg q3w achieved an objective response. Ongoing trials are evaluating bevacizumab added to CT, pembrolizumab, and tisotumab vedotin, in the phase I/II trial (NCT03786081) in recurrent patients with FIGO IVB CC [150]. The ongoing phase III trial NCT05446883 aims to assess the 2-year PFS and OS following the administration of the novel dual-targeting anti-PD-1 and anti-CTLA4 ICI QL1706 combined with CT and added or not to bevacizumab in the first-line treatment of persistent/recurrent/metastatic CC patients [131].

#### 3.2.6. Therapeutic Vaccines

The use of prophylactic HPV vaccines constitutes an important advance in the recent management of patients with CC and vaccination programs, recognized in their implementation and efficacy. Nevertheless, therapeutic vaccines are still in development. The E6 and E7 oncoproteins were expected to be used as promising targets and several novel vaccines were subsequently developed [151]. The fusion protein-based vaccine TVGV1 involving HPV16 E7 showed promising results in a mouse model, improving survival and increasing the production of HPV16 E7-specific CD8^pos^ T cells [152].

An ongoing study aims to evaluate, in the adjuvant setting, the safety and efficacy of the fusion protein vaccine involving both E6 and E7 proteins as well as the capsid HPV16 viral protein L2, involved in HPV assembly and the infectious process (TA-CIN) (NCT02405221) [153]. Another promising strategy is the combination of vaccines with immunotherapy approaches. The ongoing phase II trial VolATIL (NCT03946358) aims to evaluate the combination of UCPVax vaccine and atezolizumab in patients with locally advanced or metastatic HPV-related tumors (anal, head, neck, cervical, and vulvar cancer) [154]. CT may also contribute to enhance the efficacy of therapeutic vaccines [155]. The results of the phase I/II CervISA study [156] are, in turn, eagerly awaited regarding the safety and immune modulation effects of the HPV16 long peptide E6/E7 therapeutic vaccine (ISA101/ISA101b) in combination with carboplatin and paclitaxel with or without bevacizumab in women with recurrent or advanced HPV16-positive CC. Greater knowledge on the TME and potential synergistic effect of this new strategy with other immunomodulatory treatments need to be explored.

## 4. Perspectives

Further research is required to improve outcomes in the first-line treatment in patients with CC, currently treated with platinum-doublet CT as a standard of care. Moreover, new investigation tools are necessary and, especially, reliable biomarkers to monitor responses and better select the patients who would most benefit from immunotherapy need to be explored. Furthermore, an immunotherapy-based combination with other approaches may be a promising strategy to improve OS in these tumors, according to specific patient profiles.

Based on the recent results from the RUBY trial, further improvement in the management of relapsing dMMR EC patients is currently highly expected. Results from the ongoing trial DOMENICA, dostarlimab vs. CT in EC patients with first-line advanced or recurrent diseases are awaited. Another issue to be solved would be to better evaluate the time to relapse after last treatment, and appropriately adjust next treatments’ ICI might be an option, according to the results provided from the combination of anti-LAG-3 and anti-PD-1 leading to significant benefits in recurrent/metastatic melanoma patients. One would assume that if the LAG-3 alone or combined with the PD-1 or PD-L1 inhibitor improves the response to treatment in EC patients, then it would result in better survival. We still wonder what would be the best therapeutic option in pMMR/MSS patients. The RUBY trial showed interesting results with anti-PD-1 in the first-line setting patients. Other ongoing trials will provide elements to answer and include (i) anti-FGFR in patients with MSS metastatic EC (NCT05036681), (ii) anti-FR-α in patients with MSS advanced/recurrent EC (IMGN853), (iii) avelumab and talazoparib or axitinib in patients with MSS-recurrent/persistent EC (NCT02912572), (iv) pembrolizumab with lenvatinib in patients with pMMR (LEAP-001) [157].

Other combinations using therapeutic vaccines, but also engineered T cell transfer, with CT or RT, are also promising therapeutic approaches. In that respect, a phase I/II study (NCT05194735) evaluates the administration of autologous T-cells engineered, expressing a TCR reactive to neo-Ag in patients with relapsed/refractory solid tumors, including EC [158]. Another phase I trial has planned to enroll patients with several CD70 positive solid tumor types, including advanced/metastatic CC, and evaluate the safety and tolerability of CD70-targeted chimeric antigen receptor (CAR)-T cells (NCT05518253) [159]. Hematologic cancers greatly benefit from the development of CAR technology, and this technique is now being evaluated in patients with gynecological cancers. In parallel, bispecific antibodies targeting two immune checkpoints, such as PD-1 or CTLA-4, are currently under development, and will be tested in clinical trials enrolling patients with different cancers, including EC and CC (NCT03517488) [160,161]. Another innovative treatment strategy is the use of the bifunctional ICI, M7824, comprising the extracellular domain of the human TGF-βRII linked to the C-terminus of the human anti-PD-L1 [162]. In vivo studies showed that anti-PD-L1 reduced TGF-β signaling in the TME, decreased tumor growth, and promoted CD8^pos^ T cells’ and NK cells’ activation. The phase I/II trial (NCT04432597) tested the HPV Vaccine PRGN-2009 alone or combined with M7824 in 76 patients with HPV-positive cancers, including CC [163].

Altogether, these trials will help in the decision of the first-line treatment in patients with EC according to their MMR status, or after first recurrent disease without prior RT, CT, or surgery curative treatment. Such innovative treatment developments would benefit patients not responding to conventional therapies.

## 5. Conclusions

Until a few years ago, only limited treatment options were available in patients with advanced endometrial and cervical tumors. The increased understanding of the disease led to the emergence of new targets and therapeutic approaches. The most significant advances have been reported in the field of immunotherapy, highlighted by four pioneering studies. The basket trial Keynote-158 resulted in the approval by the FDA of pembrolizumab as a second-line treatment in cervical and endometrial cancers with PD-L1 and dMMR/MSI-H status, respectively. In patients with CC, the EMPOWER-CERVICAL-1 study is the first trial in over a decade showing survival benefits with the innovative cemiplimab, compared to the standard CT in the second-line setting, regardless of PD-L1 expression. In patients with EC, two studies marked a significant milestone leading both to treatment approval by the FDA and EMA in second-line treatment. In the GARNET trial, dostarlimab achieved a significant improvement in ORR and DOR in patients with MSI tumors. In patients with MSS tumors, the combination of pembrolizumab and lenvatinib was shown to improve DOR, PFS, and OS. Overall, all of these studies confirmed that these treatments had an acceptable toxicity profile and confirmed the already-known toxicities of immune checkpoint inhibitors.

Despite recent significant advances, tremendous progress is still expected to improve the management of EC and CC patients with an enhanced identification of predictive biomarkers for response, in both tumor and blood, to better determine the most efficient therapy and the most adequate schedule for administration.

## Figures and Tables

**Figure 1 cancers-15-02042-f001:**
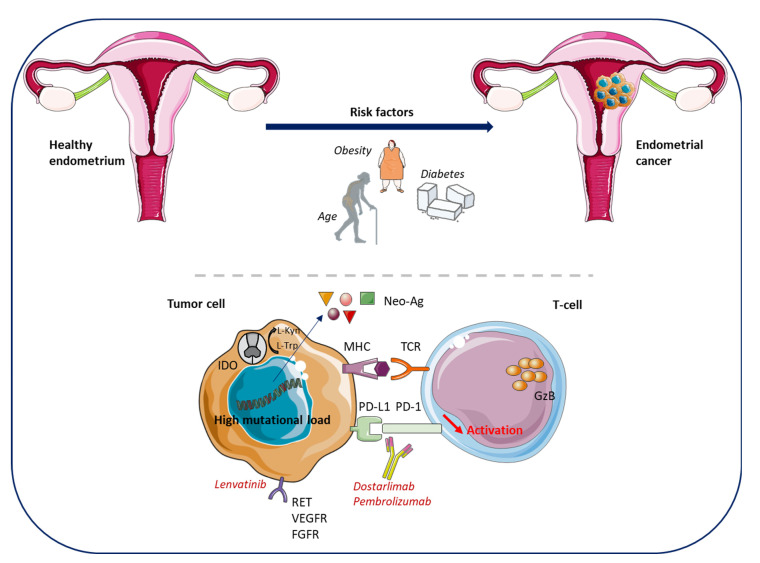
Endometrial cancer (EC)—development, immune environment and approved treatments. Risk factors linked to the development of EC include obesity, type 2 diabetes, prior breast or ovarian cancer, and age. The POLEmut and the MSI-H subtypes of EC have a high mutational load resulting in tumor-specific neo-antigen production, favoring cytotoxic T cell infiltration within tumors. EC cells are known to express IDO that converts L-Trp in L-Kyn, driving regulatory T cells’ differentiation and inhibiting CD8^pos^ and NK cells. EC cells express high levels of RET, VEGFR, and FGFR that can be targeted with Lenvatinib. EC cells also express PD-L1 and the use of Dostarlimab or Pembrolizumab, permitting the reactivation of the immune system and the anti-tumor response. Ag, antigens; EC, endometrial cancer; FGFR, fibroblast growth factor receptor; GzB, granzyme B; IDO, indoleamine 2,3-dioxygenase; Kyn, kynurenine; MSI-H, microsatellite instability hypermutated; POLEmut, polymerase ε mutation; RET, rearranged during transfection (tyrosine kinase receptor); Trp, tryptophan; VEGFR, vascular endothelial growth factor. The figure was drawn by using pictures from Servier Medical Art. Servier Medical Art by Servier is licensed under a Creative Commons Attribution 3.0 Unported License (https://creativecommons.org/licenses/by/3.0/ (accessed on 2 September 2022)).

**Figure 2 cancers-15-02042-f002:**
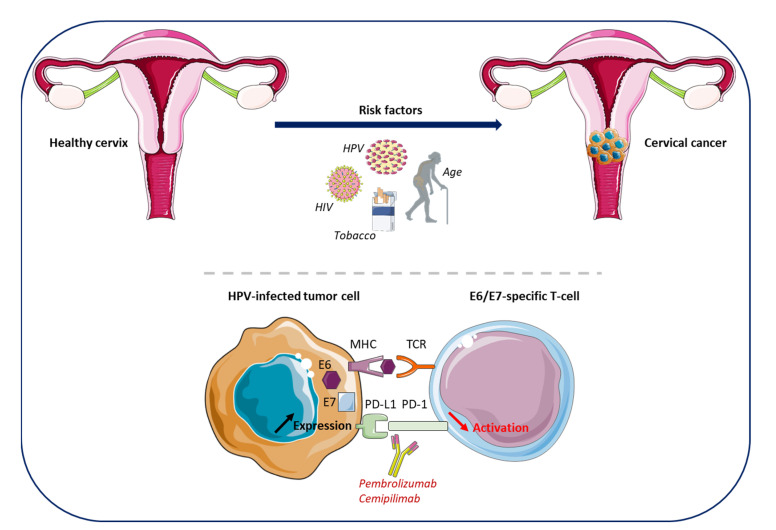
Cervical cancer (CC)—development, immune environment and approved treatments. HPV represents the major risk factor for CC development. Other risk factors include sexually transmittable co-infections such as HIV, *Chlamydia trachomatis* or genital HSV, tobacco, early age of sexual debut, multiple sexual partners, a high number of childbirths, long-term use of oral contraceptives, and age. HPV infection is highly prevalent but is eradicated in most cases by the immune system. However, a small percentage of women will develop chronic infections, resulting in the development of cancer cells that evade the immune system and, thus, leading to tumor development. One mechanism of HPV-infected cells to overcome the immune response is an increase of the PD-L1 expression at their cell surface, resulting in the inhibition of the T-lymphocyte activation through the interaction with PD-1. This inhibition can be reversed using anti-PD-1 antibodies (Pembrolizumab, Cemipilimab). CC, cervical cancer; HIV, Human Immunodeficiency Virus; HPV, Human Papilloma Virus; HSV, Herpes Simplex Virus; PD-1, programmed cell death protein 1. The figure was drawn by using pictures from Servier Medical Art. Servier Medical Art by Servier is licensed under a Creative Commons Attribution 3.0 Unported License (https://creativecommons.org/licenses/by/3.0/ (accessed on 2 September 2022)).

**Table 1 cancers-15-02042-t001:** Worldwide incidence and mortality rates of gynecological cancers in 2020.

	Worldwide New Cases [1]	Worldwide Deaths [1]	Worldwide Female Cancer Rank [1]	5-Year Survival Rates (United States)
Cervical Cancers	604,127	341,831	5th	~66% in the whole population [3] Low stage: 92% and 58% if lymph nodes are invaded [4]
Endometrial Cancers	417,367	97,370	7th	Early stages: >80% (95% for stage I) [3,5] Recurrent/advanced disease: 20-25% [5]
Ovarian Cancers	313,959	207,252	9th	~49.7% [3]
Vaginal Cancers	17,908	7,995	>10th	~49% in the whole population with a variation between 35–78% [6,7] Early stage: ~85% [7]
Vulvar Cancers	45,240	17,427	>10th	~70.3% [3]

**Table 2 cancers-15-02042-t002:** Endometrial cancer risk groups (Oaknin et al., 2022 [5]). The table can be used in stage III-IVA with complete resection without residual disease and does not apply to stage III-IVA with residual disease or for stage IV. dMMR, deficient mismatch repair; EC, endometrial cancer; G1-G3, grade 1-3; IHC, immunohistochemistry; LVSI, lymphovascular space invasion; MSI-H, microsatellite instability high/hypermutated; NSMP, no specific molecular profile; p53-abn, p53-abnormal; POLEmut, polymerase ε-ultramutated.

Risk Group	Description
Low risk	Stage IA (G1-G2) with endometrioid type (dMMR or MSI-H and NSMP) and no or focal LVSI Stage I–III POLEmut EC
Intermediate risk	Stage IA G3 with endometrioid type (dMMR and NSMP) and no or focal LVSI Stage IA non-endometrioid type (serous, clear-cell, undifferentiated carcinoma, carcinosarcoma, mixed) and/or p53-abn cancers without myometrial invasion and no or focal LVSI Stage IB (G1-G2) with endometrioid type (dMMR and NSMP) and no or focal LVSI Stage II G1 endometrioid type (dMMR and NSMP) and no or focal LVSI
High-intermediate risk	Stage I endometrioid type (dMMR and NSMP) any grade and any depth of invasion with substantial LVSI Stage IB G3 with endometrioid type (dMMR and NSMP) regardless of LVSI Stage II G1 endometrioid type (dMMR and NSMP) with substantial LVSI Stage II G2-G3 endometrioid type (dMMR and NSMP)
High risk	All stages and all histologies with p53-abn and myometrial invasion All stages with serous or undifferentiated carcinoma including carcinosarcoma with myometrial invasion All stage III and IVA with no residual tumor, regardless of histology and regardless of molecular subtype

**Table 4 cancers-15-02042-t004:** FIGO staging for cervical cancers (Cibula et al., 2018 [36]). FIGO, International Federation of Gynecology and Obstetrics grading; T, tumor; x, any number. * Pelvic sidewall is defined as the muscle, fascia, neurovascular structures, and skeletal portions of the bony pelvis.

FIGO Stage	T Category	Description
	Tx	Primary tumor cannot be assessed
	T0	No evidence of primary tumor
I	T1	Cervical carcinoma confined to the uterus (extension to corpus should be disregarded)
IA	T1a	Invasive carcinoma diagnosed only by microscope. Stromal invasion with a maximum depth of 5.0 mm measured from the base of the epithelium and a horizontal spread of 7.0 mm or less; vascular space involvement, venous or lymphatic, does not affect classification
IA1	T1a1	Measured stromal invasion of 3.0 mm or less in depth and 7.0 mm or less in horizontal spread
IA2	T1a2	Measured stromal invasion of more than 3.0 mm and not more than 5.0 mm, with a horizontal spread of 7.0 mm or less
IB	T1b	Clinically visible lesion confined to the cervix or microscopic lesion greater than IA2 (T1a2). Includes all macroscopically visible lesions, even those with superficial invasion
IB1	T1b1	Clinically visible lesion 4.0 cm or less in greatest dimension
IB2	T1b2	Clinically visible lesion more than 4.0 cm or less in greatest dimension
II	T2	Cervical carcinoma invading beyond the uterus but not to the pelvic wall or to lower third of the vagina
IIA	T2a	Tumor without parametrial invasion
IIA1	T2a1	Clinically visible lesion 4.0 cm or less in greatest dimension
IIA2	T2a2	Clinically visible lesion more than 4.0 cm in greatest dimension
IIB	T2b	Tumor with parametrial invasion
III	T3	Tumor extending to the pelvic sidewall * and/or involving the lower third of the vagina and/or causing hydronephrosis or non-functioning kidney
IIIA	T3a	Tumor involving the lower third of the vagina but not extending to the pelvic wall
IIIB	T3b	Tumor extending to the pelvic wall and/or causing hydronephrosis or nonfunctioning kidney
IVA	T4	Tumor invading the mucosa of the bladder or rectum and/or extending beyond the true pelvis (bullous edema is not sufficient to classify a tumor as T4)
IVB		Tumor invading distant organs

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
