# Peer review of "Immune Environment and Immunotherapy in Endometrial Carcinoma and Cervical Tumors"

_cancers, 2023, doi:10.3390/cancers15072042_

Round 1

Reviewer 1 Report

This is a great comprehensive review on a hot topic that requires current attention. 

Author Response

Dear Reviewer 1,

We would like to acknowledge you for the time you spent reading our manuscript. We appreciated the positive feedbacks regarding our review as well as the propositions of correction. In order to improve the English language and style, we performed a final revision of the manuscript by a medical writer.

We hope the corrections we brought to our manuscript are in line with what the reviewers were expecting. If not, we will be pleased to improve again our manuscript in order to better answer to your comments.

With kind regards,

Alexandra Lainé, PhD

Reviewer 2 Report

The MS entitled “Immune environment and immunotherapy in endometrial carcinoma and cervical tumors” is a well-planned and well-written review on the immune therapy of two gynecological cancers in their respective context of immune-ecosystem.

I enjoyed reading the MS, and I think so will the readers.

My humble suggestions are:

·       Please provide an academic reason/explanation for selecting/choosing these two gynecological neoplasms and leaving out the ovarian neoplasm in the abstract as well as in the introduction of the MS.

·       Considering the role of viral in cervical cancers (The Causal Relation between Human Papillomavirus and Cervical Cancer. J. Clin. Pathol. 2002, 55 (4), 244–265. 697 https://doi.org/10.1136/jcp.55.4.244.9 ), and its effect on the immune ecosystem, please curve out and present a separate section on this before the section on “Therapeutic vaccines.”

·       Please separately present the “Perspectives and conclusion” for the clarities shake as the perspectives tell us about our viewpoint on the topic, while the conclusion is the self-explanatory part.

·       Two diagrammatic representations of the immune environments of endometrial and cervical cancers will highly improve the presentation of this MS.

Author Response

Dear Reviewer 2,

We would like to acknowledge you for the time you spent reading our manuscript. We appreciated the positive feedbacks regarding our review as well as the propositions of correction. In order to improve the English language and style, we performed a final revision of the manuscript by a medical writer.

  • We provided an explanation for selecting endometrial/cervical gynecological cancers and not mentioning ovarian neoplasms in both abstract and introduction as follow:
    • Abstract: “However, targeting the immune system in patients with gynecological tumors remains challenging and was not always successful. In ovarian cancer, several immunotherapy treatment regimens have been investigated (as monotherapy and combination therapy in first and subsequent lines of treatment), and showed poor responses. Therefore, we specifically focalized our review on EC and CC for their specific immune-related features and therapeutic results demonstrated with immunotherapy.”
    • Introduction: Gynecological cancers include ovarian, (…)The present review will focus on uterine-associated tumors i.e. endometrial cancers (EC) (upper part, uterine corpus) and cervical cancers (CC) (lower part, uterine cervix), which showed less pronounced responses to chemotherapy (CT) than ovarian cancers, underlining the need for further therapies including immunotherapy-based treatment, in patients with EC and CC. The use of immunotherapy and in particular immune checkpoint inhibitors (ICIs) drastically changed the management and outcomes of patients, with the ability to boost endogenous immunity against unique tumor antigens resulting in long-lasting responses in some selected patients. In patients with ovarian cancer, only limited benefit was achieved, as compared with other tumors2, and partly explained because of high tumour heterogeneity and predominant immunosuppressive tumour microenvironment. Several trials currently investigated the use of ICIs in combination, with high expectation as a promising partner in innovative combined therapies. However, results with immunotherapy in patients with EC and CC are still disappointing, especially in the second line setting. Therefore, we reviewed recent achievements and suggested implementations in line with the currently expected progress.”
  • We add more evidences of the role of HPV in cervical cancers and immune-related features in sections I) Introduction/Cervical cancers and IV) Immune environment of cervical cancers
  • We separated perspectives from conclusions and it really improves the conclusion
  • We did two figures for both endometrial (Figure 1) and cervical (Figure 2) cancers representing tumor development, approved FDA/EMA drugs and main immune features regarding these cancers. They are at the end of the word file but also on a power point.

We hope the corrections we brought to our manuscript are in line with what the reviewers were expecting. If not, we will be pleased to improve again our manuscript in order to better answer to your comments.

With kind regards,

Alexandra Lainé, PhD

Reviewer 3 Report

Review

Cancers 2226189

Comments to Editors and Authors

1.       Editors,

This article is the systematic review concerning new strategy for gynecologic cancers. In gynecologic cancers, treatments for ovarian cancers were divided into platinum sensitive and platinum resistant cancers.  The majority of the tumors belong to former and nearly 10% of them belong to the latter. As for ovarian cancer has been studied well because of the good platinum sensitivity.  On the other hand, endometrial cancer (EC) and uterine cervical cancer (CC) have no striking chemotherapy and showed their remaining unchanged poor overall survivals.  So that the other therapies would be needed for EC and CC. According to development and exploit of immunotherapy, especially in immune-checkpoint-inhibitors (ICIs) using basket-clinical studies, new immunotherapeutic clinical studies have conducted.  The fundamentally, many new targets have harvested such as tumor microenvironmental interaction among CD4/8 T cells, FOXP3 positive T cell (Treg), tumor neo antigens PD1, PDL1, Leukocyte activation gene-3, Galectin-3, and so on.

In this review, up to dated new ICIs were examined precisely and the efficacy of them were demonstrated.  Further new clinical trials using new agents were presented, as well.  These knowledges would be fascinating for readers of gynecologic oncologists and useful for their new up-coming study design of clinical study.

2.       To Authors

There is misspelling in Table 4, Item IB, on line 2 of comment

T1a2/IA2 to T1a2/1a2

Please check and correct.

Author Response

Dear reviewer 3,

We first would like to acknowledge you for the time you spent reading our manuscript. We appreciated the positive feedbacks regarding our review as well as the propositions of correction. In order to improve the English language and style, we performed a final revision of the manuscript by a medical writer.

  • We corrected the misspelling in Table 4, Item IB, Line 2 with the following change: greater than IA2 (T1a2) in order to keep the FIGO and T category.

We hope the corrections we brought to our manuscript are in line with what the reviewers were expecting. If not, we will be pleased to improve again our manuscript in order to better answer to your comments.

With kind regards,

Alexandra Lainé, PhD

Reviewer 4 Report

This is a good manuscript that reviews past and current immunotherapy-based clinical trials in endometrial carcinoma and cervical tumors.

-Interpretation and presentation of trials is accurate.
-No major suggestions for improvements.

Author Response

Dear reviewer 4,

We first would like to acknowledge you for the time you spent reading our manuscript. We appreciated the positive feedbacks regarding our review as well as the propositions of correction. In order to improve the English language and style, we performed a final revision of the manuscript by a medical writer.

We hope the corrections we brought to our manuscript are in line with what the reviewers were expecting. If not, we will be pleased to improve again our manuscript in order to better answer to your comments.

With kind regards,

Alexandra Lainé, PhD
